# Plasticity versus stability across the human cortical visual connectome

Koen V. Haak [1] & Christian F. Beckmann[1,2]

Whether and how the balance between plasticity and stability varies across the brain is an important open question. Within a processing hierarchy, it is thought that plasticity is increased at higher levels of cortical processing, but direct quantitative comparisons between low- and high-level plasticity have not been made so far. Here, we address this issue for the human cortical visual system. We quantify plasticity as the complement of the heritability of resting-state functional connectivity and thereby demonstrate a non-monotonic relationship between plasticity and hierarchical level, such that plasticity decreases from early to mid-level cortex, and then increases further of the visual hierarchy. This non-monotonic relationship argues against recent theory that the balance between plasticity and stability is governed by the costs of the "coding-catastrophe", and can be explained by a concurrent decline of short-term adaptation and rise of long-term plasticity up the visual processing hierarchy.

[1] Donders Institute for Brain, Cognition and Behaviour, Radboud University Medical Center, 6500HB Nijmegen, The Netherlands. [2] Oxford Centre for Functional Magnetic Resonance Imaging of the Brain (FMRIB), University of Oxford, Oxford OX3 9DU, UK. Correspondence and requests for materials should be addressed to K.V.H. (email: k.haak@donders.ru.nl)

Cortical plasticity, the reorganization of neural circuits in response to environmental change, is ubiquitous in the brain across the lifespan. Cortical plasticity is typically considered to be beneficial because it optimizes neural processing in the face of changing environmental conditions, injury, and disease. Within processing hierarchies, however, excessive plasticity at one level of processing could disrupt the functioning of downstream neural circuits[1–9], which would require higher levels to update their interpretation of the neural code. Thus, it is important to maintain an appropriate balance between stability and plasticity.

Whether and how the balance between stability and plasticity varies across the brain is an important open question. It is thought that plasticity increases up cortical processing hierarchies[3,9,10], which would be consistent with the hypothesis that lower-level plastic changes are more costly because more dependent processing stages would have to update their interpretation of the neural code[3]. In addition, higher-order areas appear more experience dependent and they contain neurons that prefer stimuli that have been frequently encountered or that are behaviorally relevant[10–13]. The idea that plasticity is increased at higher levels of processing also agrees with observations that learning scales with task complexity, with lower-level tasks supported by lower-order areas showing less learning, and that neural changes related to learning appear larger in higher-order areas[10,14]. However, so far, no direct quantitative comparisons between lower- and higher-level plasticity have been made. Here, we addressed this issue for the human cortical visual system.

Determining whether and how plasticity varies across the visual processing hierarchy requires a quantifiable measurement process. One possible approach would be to characterize the configuration of the neural circuitry at one point in time and then determine how much it changed at a later point. This approach requires a longitudinal study design. Another, equally valid approach is to assess the current state of configuration with respect to a state where the configuration was free of environmental influence. Such a state of zero-change corresponds to the configuration of neural circuits that is completely determined by the genetic blueprint, and hence plasticity can be quantified as the complement of the amount of phenotypic variance that can be explained by genetic factors. The amount of phenotypic variance that can be explained by genetic factors is known as heritability, which can be estimated under a twin study design. In the present work, we adopted the latter approach, as it allowed us to gauge the totality of plastic changes that occurred across the entire lifespan up to the time of measurement, and because it enabled answering our research question based on the publicly available neuroimaging data of the WU-Minn Human Connectome Project[15].

There are several possible phenotypes that can be assessed. For instance, plastic changes might be assessed in terms of stimulus and/or task-related neural responses. However, it is often unclear what stimulus or task should be used to target specific processing levels, and under external stimulation and/or a task it would be difficult to distinguish true variations in plasticity from constant plasticity with differences in expression due to limitations imposed by neuronal response properties specific to each processing level[9]. In addition, although for primary processing nodes (e.g., the retina of the eye) plasticity may be defined as a change in response to external stimulation, for higher-order processing nodes it ought to be defined as a change in response to the signals these nodes receive from lower-level processing stages (i.e., because response changes at higher levels could be a manifestation of lower-level plastic changes). Plastic changes might also be assessed in terms of anatomical features. However, plasticity is not limited to anatomical changes, as evidenced by, for instance,

adaptive neural tuning changes in response temporarily altered image statistics[8] and the existence of long-term potentiation (LTP) to facilitate learning and memory by synaptic strengthening[16].

Given these considerations, we elected to estimate the amount of plasticity across the human cortical visual system based on resting-state functional connectivity (RSFC): the temporal correlations between spontaneous functional MRI signal fluctuations at different cortical sites that arise in the absence of an explicit task or stimulus[17]. RSFC has been shown to predict inter-individual differences in stimulus- and task-evoked brain activity in multiple behavioral domains, indicating that it respects the functional interactions seen under perception, action, and cognition[18]. However, because it is determined with no stimulus or task, it avoids the issues with assessing plasticity in terms of stimulus- and/or task-related responses. In addition, though RSFC adheres closely to anatomical connectivity[19–21], it is a measure of function and hence not limited to capturing only anatomical changes. Thus, by using RSFC as our phenotype of interest, we avoided not only the issues associated with assessing plasticity across brain areas in terms of stimulus- and/or task-related neural responses, but also the limitations associated with purely anatomical features.

## Results

**RSFC heritability across visual cortex.** To quantify the balance between plasticity and stability across human visual cortex, we determined the functional connectivity between 48 cortical visual areas based on the publicly available resting-state functional MRI (rfMRI) data of twins provided by the WU-Minn Human Connectome Project (HCP)[15,22]. Our sample included only those subjects whose twin-status was genetically confirmed, and who had completed all rfMRI sessions. Thus, our sample consisted of 123 monozygotic (MZ) and 67 dizygotic (DZ) twin-pairs (380 subjects in total).

Visual areas were delineated using a publicly available atlas of human retinotopic cortex[23], and we extracted the average rfMRI time-series from each of them. For each subject, we regressed out the effects of head-movement and computed functional connectivity between each area-pair as the correlation between the residual time-series. The statistical significance of each connection was assessed by block-permutation[24], and all estimates whose FDR-corrected $p$-value exceeded 0.05 were excluded from further analysis. The heritability ($h^2$) of the functional connectivity between all possible area-pairs was estimated with covariates age, sex, MR reconstruction software version, and a summary measure of each subject's head motion during scanning (i.e., mean frame-wise displacement). Figure 1a shows the estimates for each statistically significant ($p < 0.05$, FDR corrected) connection.

We further ascertained that the heritability estimates could not be explained by possible confounding effects. For instance, visual areas differ in size and the temporal signal-to-noise ratio also varied across the occipital lobe. This could have led to differences in measurement error, which could in turn have influenced the heritability estimates. However, this was not the case, because the test–retest reliability of the connectivity estimates did not covary with heritability (Mantel test: $R^2 = 0.01$, $p = 0.33$). In addition, atlas-based area definitions are not expected to fit equally perfectly across subjects, which could have influenced the heritability estimates because this misfit is likely heritable in its own right[25]. However, this was also not the case, because the heritability estimates were unrelated to the expected precision of area definition ($F$-test, $R^2 = 0.02$, $F_{1,44} = 1.07$, $p = 0.31$)—see Methods for details.

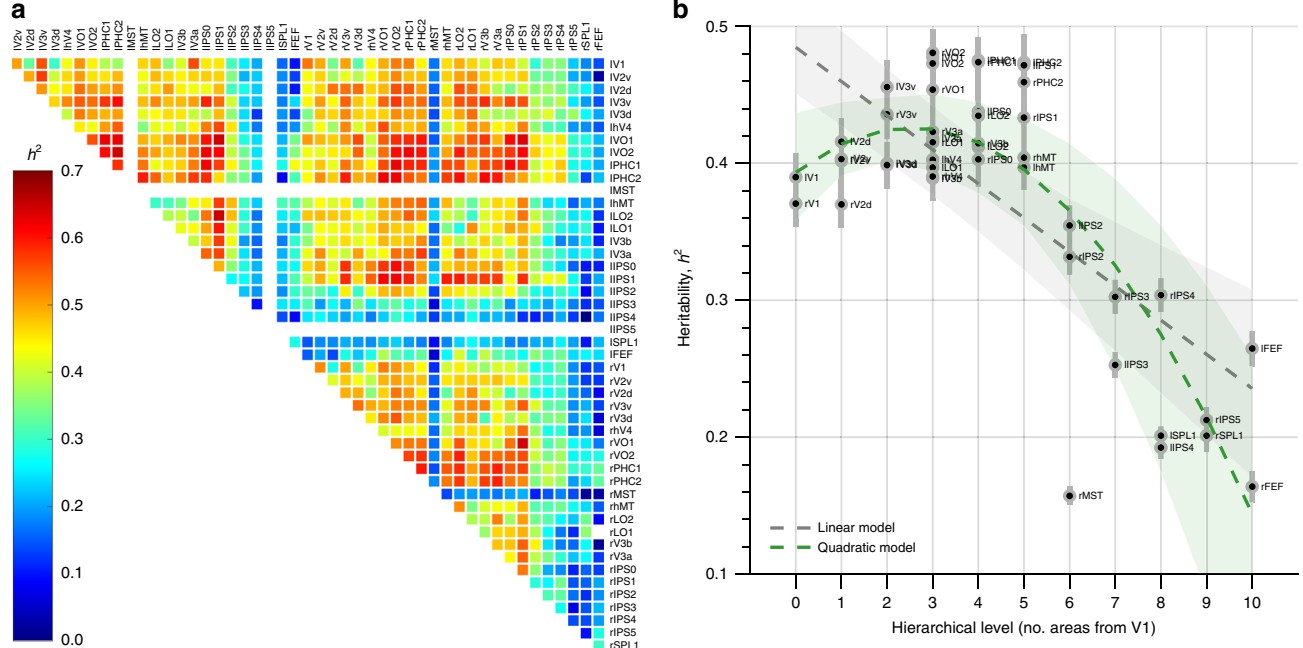

**Fig. 1** Heritability of functional connectivity across human visual cortex. **a** Heritability estimates for all significant ($p < 0.05$, FDR corrected) functional connections. Visual area names are abbreviated according to Wang et al. (2015). **b** Average heritability of functional connectivity as a function of hierarchical level. Gray error-bars indicate the SEM. Dashed gray and green lines and similarly colored shaded areas indicate linear and quadratic model fits and the bootstrapped 95% confidence intervals, respectively. Source data are provided as a Source Data file

**RSFC heritability as a function of hierarchical level**. Next, we asked whether the heritability of each area's connectivity 'fingerprint' was related its hierarchical level. To this end, we computed for each visual area the average heritability across all of its connections. As in previous work[26], we determined each area's hierarchical level as the number of visual areas between that area and area V1. Ideally, the hierarchical level of a visual area is determined by assessing the laminar origin and termination patterns of their connections[27]. However, such information is not yet available for humans and the present approximation corresponds well to a data-driven estimation of hierarchical level based on multidimensional scaling[26], which also correctly determined hierarchical level from a matrix based on the laminar origin and termination patterns in the macaque[28].

Figure 1b shows the heritability of each area's connectivity fingerprint as a function of hierarchical level. To test if heritability was significantly related to hierarchical level, we performed linear and quadratic regression analyses. Whilst both models indicated a highly significant relation between heritability and hierarchical level ($F$-tests; linear: $R^2 = 0.49$, $F_{1,44} = 43$, $p = 5.1 \times 10^{-8}$; quadratic: $R^2 = 0.67$, $F_{2,43} = 44.1$, $p = 3.8 \times 10^{-11}$), the quadratic model fitted the data better than the linear model ($F$-test, $F_{1,43} = 23.3$, $p = 1.8 \times 10^{-5}$) because heritability increased significantly from V1 to V3 ($t$-test, $t_8 = 2.38$, $p = 0.04$). Thus, plasticity (i.e., the complement of heritability) is indeed greater at higher versus lower levels of visual processing, but it does not increase monotonically up the visual processing hierarchy.

**Short-term and long-term plasticity components**. The non-monotonic relationship between RSFC heritability and hierarchical level may be explained assuming that RSFC heritability is influenced by both short-term and long-term plastic processes. Indeed, RSFC heritability is likely influenced by sustained long-term as well as transient short-term plasticity. These two components would be expected to respectively rise and fall as a

function of hierarchical level such that the net amount of plasticity follows the observed non-monotonic relationship (Fig. 2). To add weight to the premises of this model—namely, that short-term plasticity decreases and that long-term plasticity increases as a function of hierarchical level—we tested (1) whether the test–retest reliability of the RSFC estimates across two consecutive session days increases with hierarchical level, and (2) whether the heritability of anatomical phenotypes decreases up the hierarchy. That is, presumably, transient differences in functional connectivity can be due to short-term but not long-term plastic processes, whereas anatomical phenotypes may be influenced by sustained long-term but not transient short-term plastic changes.

In the preceding test–retest reliability analysis, we leveraged the test–retest reliability of the functional connectivity estimates to confirm that measurement errors did not influence the heritability estimates at the level of individual connections. However, this does not imply that the test–retest reliability does not increase with hierarchical level. In the current analysis, therefore, we determined whether the RSFC test–retest reliability was related to hierarchical level, averaging for each visual area the connection-specific test–retest reliability estimates across all of its connections. Figure 3 shows that the test–retest reliability indeed increases significantly with hierarchical level ($t$-test, $t_{44} = 2.72$, $p = 9.2 \times 10^{-3}$). In addition, the test–retest reliability was more likely associated with the short-term than the long-term plasticity component derived from fitting the full model to the RSFC heritability data shown in Fig. 2b (relative likelihood of short-term vs. long-term model = 1; short-term and long-term component model fitting involved estimating an intercept $\beta$ as a single free parameter; the scale and exponential decay of each component were fixed as the parameter estimates obtained by fitting the full two-component model against the RSFC heritability estimates; i.e., the red and blue curves in Fig. 2b were only allowed to move up and down). This result not only adds weight to the premise that short-term plasticity decreases up the visual hierarchy, but also rules out that the overall decline of

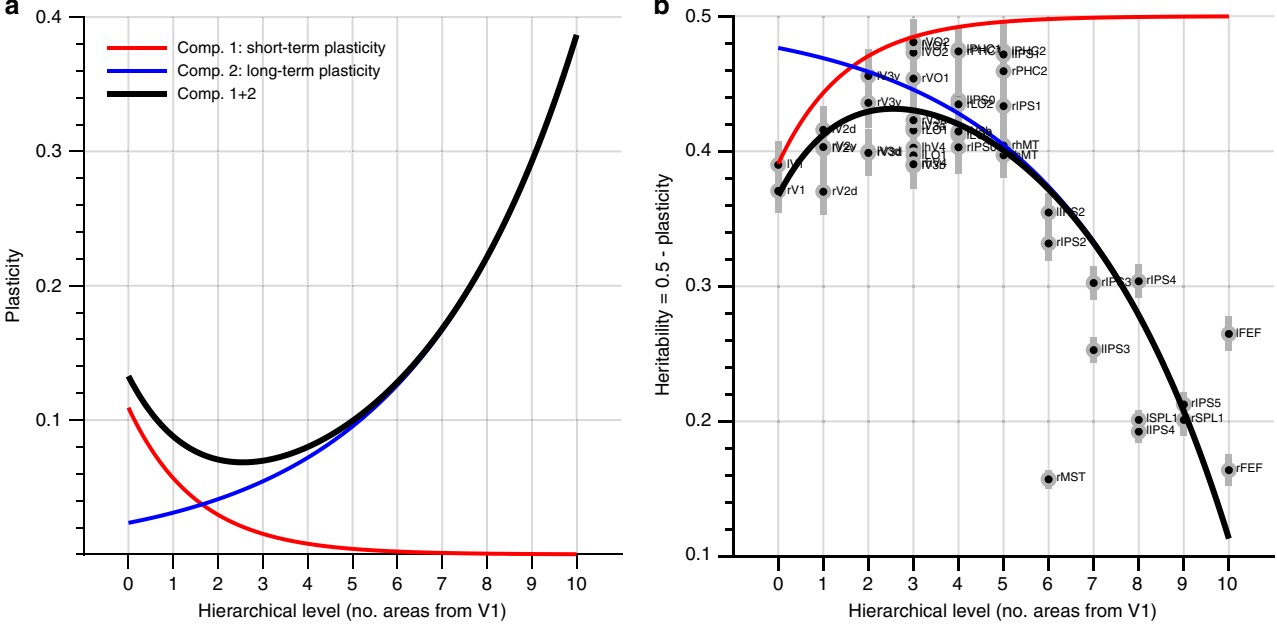

**Fig. 2** Theoretical model of the relationship between heritability and hierarchical level. **a** The non-monotonic relationship between heritability and hierarchical level can be modeled as the complement of the sum of transient short-term plasticity and sustained long-term plasticity components. Under this model, short-term plasticity (red) is relatively weak and decreases exponentially from low to high levels of visual processing, whereas long-term plasticity (blue) increases exponentially to become relatively strong in high-level visual cortex. **b** Under the assumption that the average heritability estimates shown in Fig. 1b peak at 0.5 (due to e.g. measurement noise common to all hierarchical levels of processing), the complement of the sum of both components (black) predicts the observed heritability estimates very well ($R^2 = 0.73$). Source data are provided as a Source Data file

RSFC heritability can be explained by increasing measurement error up the visual processing hierarchy.

To add weight to the second premise that long-term plasticity increases up the visual hierarchy, we estimated—for each visual area—the heritability of its cortical thickness and gray-matter volume. Here, the assumption is that sustained changes in functional connectivity are observable as changes in anatomical features. These anatomical phenotypes are presumably free of short-term plastic changes and should therefore exhibit a monotonic decrease in heritability. Figure 4 shows that this indeed is the case. The heritability of cortical thickness and gray-matter volume was negatively related to hierarchical level (t-tests; thickness: $t_{44} = -3.42$, $p = 1.4 \times 10^{-3}$; volume: $t_{44} = -3.62$, $p = 7.5 \times 10^{-4}$) and did not increase from V1 to V3 (t-tests; thickness: $t_8 = -0.08$, $p = 0.93$; volume: $t_8 = -3.34$, $p = 0.01$). In addition, and in contrast to the test–retest reliability of the functional connectivity estimates, the heritability of both anatomical phenotypes was more likely associated with the long-term than the short-term plasticity component shown in Fig. 2 (relative likelihood of short-term vs. long-term model <0.01). Taken together, these results conform to our two-component model of the expression of short-term and long-term plasticity across the visual hierarchy.

## Discussion
We investigated whether and how the plasticity of functional connectivity varies across the cortical visual hierarchy. To this end, we leveraged the fact that plasticity can be quantified as the complement of heritability. That is, we quantified plasticity as the amount of deviation from the genetic blueprint. The results indicate a non-monotonic relationship between plasticity and hierarchical level, such that plasticity decreases from V1 to V3 and then increases further up the visual hierarchy.

The observation that plasticity decreases from V1 to V3 indicates that the first stages of cortical visual processing are not the most stable. This challenges theory that the amount of plasticity depends on a neural circuit's hierarchical level because plastic changes at one level cause cascading effects downstream where higher-level circuits need to update their interpretation of the new neural code[1–4,6]. This "coding catastrophe" confers lower costs to plasticity at higher levels of the processing hierarchy, because higher levels have fewer downstream dependents. If these costs critically determined the amount of plasticity along cortical processing hierarchies, plasticity should monotonically increase up the visual hierarchy, and the first stage of cortical processing (V1) should be the most stable. However, the present data suggest that this is not the case.

Why plasticity is distributed this way is an important open question. One possibility is that plasticity decreases along the ventral occipital surface, whereas it increases up the lateral and dorsal occipital surfaces. Indeed, areas along the ventral occipital surface exhibited greater heritability than areas on the lateral and dorsal occipital surfaces (Fig. 1b), and previous work also noted differences in ventral versus dorsal visual plasticity[9,29]. This account raises the question whether it be possible that higher dorsal visual areas are more plastic because they are increasingly involved in interacting with the environment, whereas higher ventral areas are less plastic because they implement more stable integration codes (e.g., to enable object recognition under various viewing conditions) that are robust against the coding catastrophe.

Another possibility is that the increasing heritability from V1 to V3 reflects decreasing transient adaptive changes in response to recent visual experience, whereas the overall decline in heritability reflects a rise in more permanent plastic changes due to learning. Indeed, different types of plasticity may be distinctly expressed at different cortical locations, and it is therefore

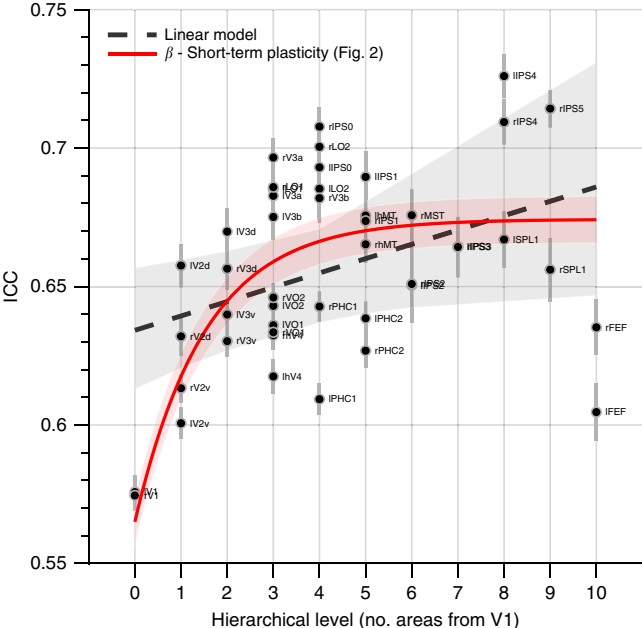

**Fig. 3** RSFC differences (ICC) across session days as function of hierarchical level. Error-bars indicate the SEM. The dashed gray line and gray-shaded area indicate linear model fits and the bootstrapped 95% confidence interval, respectively. The positive relation between ICC and hierarchical level ($t$-test, $t_{44} = 2.72$, $p = 9.2 \times 10^{-3}$) indicates smaller differences in higher-level visual areas, which is consistent with decreased effects of short-term plasticity up the visual hierarchy predicted by the theoretical model presented in Fig. 2. The thick red line and red shaded area indicate the fit of the short-term plasticity component and its 95% confidence interval (only the intercept was fitted, the other parameters were fixed as the estimates based on fitting the full two-component model to the RSFC heritability data shown in Fig. 2b). Source data are provided as a Source Data file

possible that short-term plasticity is most pronounced in V1 and then decreases up the visual hierarchy, while long-term plasticity is increasingly pronounced in higher-level visual areas (Fig. 2). This account agrees with both a lack of long-term V1 plasticity after retinal damage[1,30–32] and the large body of evidence of short-term adaptation[8] in early visual cortex. The account is also in line with smaller differences between the functional connectivity estimates across consecutive testing days at higher levels of visual processing (Fig. 3)—which presumably are unrelated to long-term plastic changes—and a monotonically decreasing heritability up the hierarchy when heritability is estimated based on purely anatomical phenotypes (Fig. 4)—which presumably are free of short-term plastic changes.

Importantly, these accounts are not mutually exclusive because visual processing along the ventral occipital surface might be primarily governed by short-term plastic changes—indeed, the red line in Fig. 2b adheres closely to the heritability estimates for the ventral visual areas—whereas visual processing along the lateral and dorsal occipital surfaces might be governed by the sum of short and long-term plasticity (thick black line in Fig. 2b).

By quantifying plasticity as the complement of heritability, our measure covers the totality of all possible changes with respect to the genetic blueprint across the entire lifespan up to the time of measurement. This means that we cannot comment on whether these changes occurred during childhood or adulthood. It is further possible that some of the changes we attribute to plasticity are in fact non-plastic changes that occurred due to ageing, injury, or disease. However, because our sample included only healthy

young adults in whom the visual system should be fully developed but not yet aged[33], the contribution of such non-plastic changes to the overall amount of change should be negligible.

In conclusion, our results indicate that the notion of increased plasticity at higher levels of cortical processing is an over-simplification. Rather, they suggest that there are different types of plasticity (e.g., short- and long-term plasticity), the expression of which varies distinctly across the cortical visual system. This principle may apply across the sensory modalities and species. Although our results pertain principally to plastic changes under normal conditions, they may also have clinical relevance when used to gauge how much plasticity can be expected in response to focal brain injuries and neurological disease.

## Methods

**Dataset and pre-processing**. The dataset comprised subjects of the S1200 WU-Minn HCP[15] data-release who completed all of the four rfMRI runs ($4 \times 1200$ time points) and whose twin-status was confirmed by genetic testing. The dataset included 123 monozygotic (MZ) and 67 dizygotic (DZ) twin-pairs. All participants provided written informed consent. Subject recruitment procedures and informed consent forms, including consent to share de-identified data, were approved by the Washington University in St. Louis Institutional Review Board (IRB). The 2 mm isotropic, multiband accelerated (x8) 3T rfMRI data with a TR of 0.72 s were pre-processed as detailed in Smith et al.[22], which involved rigorous data cleaning using the FIX artefact removal procedure[34,35]. For the present work, we additionally smoothed the images using a 3-mm FWHM Gaussian kernel.

**Regions-of-interest definition**. Visual areas were defined using a probabilistic atlas[23]. The atlas provides both full probability maps and maximum probability maps (i.e., the most probable area label for any given point) in MNI space. We used the latter for region-of-interest definition and down-sampled the region definitions from 1 mm to 2 mm isotropic resolution using nearest-neighbor interpolation. Furthermore, a single V1 region was defined by combining the maximum probability maps labeled V1d and V1v in each hemisphere. As such, the number of regions-of-interest that were used in subsequent analysis steps was 48 (24 in each cerebral hemisphere).

**Functional connectivity analysis**. We determined the functional connectivity between all possible area-pairs at the individual level by (1) extracting the average time-series from each region-of-interest, (2) regressing out the motion realignment parameters and their first derivatives, and (3) computing the (Fisher's $r$-to-$z$ transformed) correlation between the residuals. Next, we determined the group-level statistical significance of the functional connectivity estimates using a block-permutation test[24] to account for the family structure in the dataset. The ensuing $p$-values were corrected for multiple comparisons using a false-discovery rate (FDR) approach, and all connections whose FDR-corrected $p$-value exceeded 0.05 were excluded from further analysis.

**Heritability analysis**. The heritability of each connection was estimated using the freely available SOLAR software package[36]. Heritability, which is defined as the portion of phenotypic variance accounted for by the total additive genetic variance (i.e. $h^2 = \sigma_g^2/\sigma_p^2$), was assessed with simultaneous estimation for the effects of covariates age, sex, a measure of each subject's head motion during scanning (mean frame-wise displacement), and MRI reconstruction software version. SOLAR estimates the variances $\sigma_g^2$ and $\sigma_e^2$ by comparing the observed phenotypic covariance with the covariance predicted by kinship (i.e. $\Omega = 2\Phi\sigma_g^2 + I\sigma_e^2$, where $\Phi$ is the kinship matrix), and determines the statistical significance of the heritability estimates by comparing the log-likelihood of the model in which $\sigma_g^2$ is constrained to be zero ($L_0$) to the log-likelihood of the model in which $\sigma_g^2$ is estimated ($L_e$). This is done using the test-statistic $2(L_e - L_0)$ which is asymptotically distributed as a 50:50 mixture of a $X_0^2$ (point mass) and a $X_1^2$ distribution. Prior to analysis, the functional connectivity estimates were subjected to the inverse normal transformation to ensure that the residual kurtosis (i.e. the kurtosis after the covariate influences have been removed) was within normal range.

Given that both the size and temporal signal-to-noise ratio (tSNR) of the visual areas varied across the occipital lobe, it was important to rule out that the heritability estimates were biased by measurement error. Therefore, we leveraged the fact that each subject was scanned on two different session days (each session day involved 2400 time points; i.e. two rfMRI runs of 1200 MR volumes) and re-computed the functional connectivity matrix for each session day to determine the test–retest reliability. If the heritability estimates co-varied with measurement error, this would result in a correlation between the test–retest reliability across connections and heritability. Thus, we computed the intra-class correlation coefficient ($ICC_{3,1}$)[37,38] for each connection and then determined the variance in the heritability matrix (Fig. 1a) that could be explained by the ICC matrix. We used

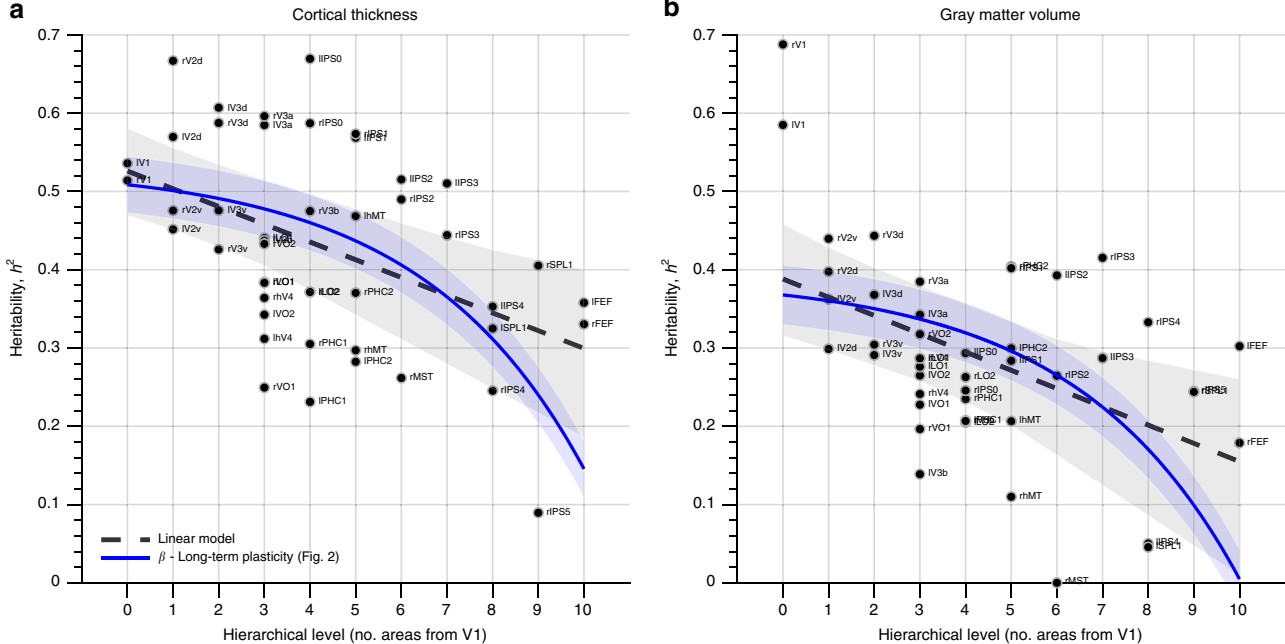

**Fig. 4** Heritability of two anatomical phenotypes as a function of hierarchical level. The dashed gray lines and shaded areas indicate linear model fits and 95% bootstrapped confidence intervals, respectively. **a** The heritability of cortical thickness was negatively related to hierarchical level ($t$-test, $t_{44} = -3.42$, $p = 1.4 \times 10^{-3}$) and there was no significant increase in heritability from V1 to V3 ($t$-test, $t_8 = -0.08$, $p = 0.93$). **b** The heritability of gray-matter volume was also monotonically and negatively related to hierarchical level ($t$-tests; across all areas: $t_{44} = -3.62$, $p = 7.5 \times 10^{-4}$; from V1 to V3: $t_8 = -3.34$, $p = 0.01$). These results are consistent with the theoretical model presented in Fig. 2, because purely anatomical phenotypes are presumably not strongly influenced by short-term plasticity. The thick blue lines and blue shaded areas indicate the fits of the long-term plasticity component and 95% confidence intervals (only the intercept was fitted, the other parameters were fixed as the estimates based on fitting the full two-component model to the RSFC heritability data shown in Fig. 2b). Source data are provided as a Source Data file

a non-parametric permutation approach to test for statistical significance (i.e. a Mantel test with 5000 permutations), because the values within either matrix are not independent.

In addition, we ascertained that the heritability estimates shown in Fig. 1b were not related to the expected precision of area definition. This was important because atlas-based area definitions are not expected to fit equally perfectly across subjects, which could have influenced the heritability estimates. To verify that this was not the case, we regressed the likelihood that cortical points in a given area would be classified as part of that area in unseen subjects (i.e., the mean of the corresponding probability map masked with the atlas definition of that area) onto the heritability estimates (averaged across all of an area's connections because we only have one estimate of area definition precision per area), and determined statistical significance using an $F$-test.

**Regression analyses**. To determine the relation between the heritability of functional connectivity and hierarchical level, we first summarized the heritability of each area's connections by averaging the heritability estimates across all of that area's connections. By doing so, we effectively assess the heritability of an area's entire functional connectivity profile with respect to the rest of visual cortex (i.e., its functional connectivity fingerprint), with the added benefit that there is no need to further correct for effects of distance because the average distance from one area to all other areas is equal for all areas. As in previous work[26], we determined the hierarchical level of each visual area by constructing a nearest-neighbor graph with edges between areas only if they are direct neighbors. We then used Dijkstra's algorithm to determine the shortest path through this graph from each visual area to V1. The length of this shortest path (i.e., the number of areas that need to be visited before V1 can be reached) was our measure of hierarchical level.

We considered two possible relationships between heritability and hierarchical level: (1) a linear model (i.e., $\bar{h}^2 = \beta_0 + \beta_1 \eta + \varepsilon$, where $\eta$ represents hierarchical level) in accordance with the hypothesis that heritability decreases monotonically up the visual hierarchy, and (2) a quadratic model (i.e., $\bar{h}^2 = \beta_0 + \beta_1 \eta + \beta_2 \eta^2 + \varepsilon$), because visual inspection of the data strongly suggested that the heritability of early visual connectivity was much lower than might be expected had the relation been a strictly monotonic linear decrease. The significance of the two models was assessed by $F$-tests and the two (nested) models were compared using a $F$-ratio test. Finally, we determined the relationship between heritability and hierarchical level within early visual cortex (V1, V2d, V2v, V3d, and V3d in both hemispheres; 10 areas

total) by linear regression, and assessed the statistical significance of the slope (two-sided $t$-test).

**Two-component model of plasticity**. To model the observed relationship between heritability and hierarchical level, we assumed that heritability peaks at 0.5 (to account for unbiased measurement noise common to all stages of visual processing) and that total plasticity can be decomposed as the sum of two components: one of which is relatively weak and decreases exponentially with hierarchical level (i.e. $c_1 = a_1 \cdot e^{-b_1 \eta}$, where $\eta$ is hierarchical level), and one that is relatively strong and increases exponentially with hierarchical level (i.e. $c_2 = a_2 \cdot e^{b_2 \eta}$). As such, the observed heritability is predicted by $h^2 = 0.5 - (c_1 + c_2)$. Parameters $a_1, b_1, a_2,$ and $b_2$ were estimated using robust non-linear least-squares regression.

To test if $c_1$ might reflect short-term plasticity, we determined the relationship between ICC (averaged per area) and hierarchical level. To test if $c_2$ might reflect long-term plasticity, we tested whether the heritability of two purely anatomical phenotypes (see "Heritability of anatomical phenotypes" below) was related to hierarchical level. These relationships were determined by linear regression and the statistical significance of the slope was assessed by a two-sided $t$-test. For the anatomical phenotypes we also tested whether their heritability decreased with hierarchical level in early visual cortex (i.e., V1, V2d, V2v, V3d, and V3d in both hemispheres; 10 areas total) to ascertain that the initial increase in heritability observed for the functional connectivity estimates was absent for purely anatomical features. In addition, we determined the relative likelihood of each model component by fixing $c_1$ and $c_2$ (i.e., parameters $a_1, b_1, a_2,$ and $b_2$ were fixed based on fitting the full model to the heritability of functional connectivity) and then fitting $\beta_1 - c_1$ and $\beta_2 - c_2$ to the ICC and anatomical heritability data. This was done using non-linear least-squares regression with $\beta_1$ and $\beta_2$ as single free parameters, after which the relative likelihood was determined as the Akaike weight for each component model.

**Heritability of anatomical phenotypes**. We considered two anatomical phenotypes: cortical thickness and gray-matter volume. Individualized estimates of the cortical thickness for each visual area were obtained by computing the average cortical thickness within that area, where cortical thickness refers to the raw thickness estimates from the HCP FreeSurfer pipeline[39] corrected for linear effects of curvature (this latter step is important when the interest in comparisons across cortical areas because surface folding makes gyri thicker and sulci thinner). Estimates of gray-matter volume were obtained for each individual using FSL-VBM[40]

with standard settings applied to each subject's bias-field corrected T1-weighted anatomical. These voxel-wise estimates were spatially smoothed using a Gaussian kernel with a FWHM of 3 mm (note that this amount of smoothing is lower than is typical for voxel-based morphometry analyses because the interest is in local estimates of gray-matter volume within small brain areas) and averaged per area to obtain a single estimate of gray-matter volume per subject for each area. The heritability of each area's cortical thickness and gray-matter volume was estimated with covariates age and sex by the procedures described above.

**Reporting summary**. Further information on research design is available in the Nature Research Reporting Summary linked to this article.

## Data availability
The source data underlying Figs. 1, 2, 3, and 4 are provided as a Source Data file. All relevant MRI data are publicly available at https://db.humanconnectome.org. Information about the family structure in the HCP data is available at https://db.humanconnectome.org to qualified investigators who agreed to HCP's Restricted Data Use Terms.

## Code availability
The heritability analyses require information about the family structure in the HCP data, which is restricted by the HCP due to legal and ethical issues pertaining to confidentiality and privacy of participants. The code for performing these analyses is therefore available from the corresponding author after providing proof of access to HCP restricted data. All other analysis code is readily available from the corresponding author upon reasonable request.

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

## Acknowledgements
This work was supported by the Netherlands Organization for Scientific Research Veni Grant No. 016.Veni.171.068 (to K.V.H.), and Vidi Grant No. 864-12-003 (to C.F.B.). Data were provided by the Human Connectome Project, WU-Minn Consortium (Principal Investigators: David Van Essen and Kamil Ugurbil; 1U54MH091657) funded by the 16 NIH Institutes and Centers that support the NIH Blueprint for Neuroscience Research; and by the McDonnell Center for Systems Neuroscience at Washington University.

## Author contributions
K.V.H. and C.F.B. conceptualized the research, analyzed the data and wrote the manuscript.

## Additional information

**Competing interests:** C.F.B. is director and shareholder of SBGneuro Ltd. The remaining author declares no competing interests.

