## [Transparent Peer Review File · Nature Communications]

Reviewers' comments:

Reviewer #1 (Remarks to the Author):

This manuscript describes analyses using HCP data measuring plasticity of the visual system. The authors used the HCP twin data to calculate heritability of functional connectivity between pairs of visual cortical regions (i.e., long-term plasticity across the subjects' lives to that point). Despite expectations based on prior literature and theory that plasticity increases with increasing level of the visual hierarchy, the authors observed an initial decrease in plasticity from primary visual cortex to mid-level visual regions, before a large increase in plasticity between mid-level visual regions and higher-level visual regions. This finding may be explained by differentiating between short-term adaptation in primary visual cortex and longer-term plasticity in higher-level visual regions.

This manuscript addresses an important and relevant topic, and does an excellent job at setting up the topic, describing the analyses and results, and discussing the unexpected findings. I have only two minor comments:

First, more of a discussion in the introduction regarding why functional connectivity is a relevant outcome variable would be useful to justify your choice. You also don't explicitly mention that you are using resting state data in the introduction (although you do allude to it). Since HCP data also has task data where you could look at stimulus-evoked visual system connectivity in addition to resting state connectivity, more elaboration on your choice to use only resting state data would be useful as well.

Second, since you have two sessions of data for each participant, you could empirically test whether short-term plasticity is higher in primary visual regions and decreases as you move up the visual hierarchy. Presumably, long-term plasticity wouldn't change much between the two sessions for an adult participant. Short-term plasticity would be more likely to change, given the sessions are on different days. Therefore, a greater difference across testing sessions would more likely reflect short-term than long-term plasticity. Do you see reduced variability across sessions within individuals in the functional connectivity profiles of higher-level visual regions? This would be an interesting analysis to add to your manuscript.

Reviewer #2 (Remarks to the Author):

- The key idea of this letter is to quantify plasticity as the Complement of the Heritability of Functional Connectivity (CHFC). This is an important and natural idea, but it raises the question - discussed by the authors- of whether functional connectivity (which relates only to correlation) is the right or the only measure that should be considered to assess plasticity in this specific dataset. As mentioned, plasticity may also be assessed using phenotypic variation with time, or longitudinal changes in brain structure. It would be interesting - and not too difficult - to evaluate if anatomical phenotypes (e.g. grey matter volume or surface area) concur with the proposed theory.

- The methodological caveat that heritability, defined as the proportion of variance explained by genotypic information over the total phenotypic variance, is dependent on the measurement noise level is well accounted for. Functional connectivity measures could indeed vary with noise level and this can be measured with test re-test reliability, although it is not clear how many time point measures were used for this assessment. Other measures of the noise level per participant could be obtained, and the normalized effect sizes across the visual hierarchy estimated.

- The discussion on how the overall plasticity could be divided into short term (ventral stream) and long term (dorsal stream) plasticity components is interesting and fits with some of the literature. It is indeed plausible that the relative amount of plasticity measured by CHFC may be governed by several factors. It also begs the question whether it would be possible that the higher level of hierarchy in the ventral stream could correspond to more stable integration neural codes and therefore may be less subject to the coding catastrophe theory.

- It might be worth trying to -at least in part- test the theory in other sensory cortices (e.g. auditory).

- Overall, I have found this a fascinating article and I congratulate the authors for their work: the estimation of plasticity they propose seems to be an excellent idea, and their explanation on how this informs the balance between plasticity and stability in the visual cortex is compelling.

Response to Reviews

We thank the editor and reviewers for their time and positive evaluation of our work. We are also grateful for their insightful comments and suggestions, most of which we have now implemented in the revision to further improve the manuscript. Please find our response to each comment below. All changes have been highlighted in green in the manuscript file.

Reviewer #1 (Remarks to the Author):

This manuscript describes analyses using HCP data measuring plasticity of the visual system. The authors used the HCP twin data to calculate heritability of functional connectivity between pairs of visual cortical regions (i.e., long-term plasticity across the subjects' lives to that point). Despite expectations based on prior literature and theory that plasticity increases with increasing level of the visual hierarchy, the authors observed an initial decrease in plasticity from primary visual cortex to mid-level visual regions, before a large increase in plasticity between mid-level visual regions and higher-level visual regions. This finding may be explained by differentiating between short-term adaptation in primary visual cortex and longer-term plasticity in higher-level visual regions.

This manuscript addresses an important and relevant topic, and does an excellent job at setting up the topic, describing the analyses and results, and discussing the unexpected findings. I have only two minor comments:

First, more of a discussion in the introduction regarding why functional connectivity is a relevant outcome variable would be useful to justify your choice. You also don't explicitly mention that you are using resting state data in the introduction (although you do allude to it). Since HCP data also has task data where you could look at stimulus-evoked visual system connectivity in addition to resting state connectivity, more elaboration on your choice to use only resting state data would be useful as well.

Response:

We thank the reviewer for this suggestion and have revised the introduction to further clarify our motivation for using resting-state functional connectivity.

Second, since you have two sessions of data for each participant, you could empirically test whether short-term plasticity is higher in primary visual regions and decreases as you move up the visual hierarchy. Presumably, long-term plasticity wouldn't change much between the two sessions for an adult participant. Short-term plasticity would be more likely to change, given the sessions are on different days. Therefore, a greater difference across testing sessions would more likely reflect short-term than long-term plasticity. Do you see reduced variability across sessions within individuals in the functional connectivity profiles of higher-level visual regions? This would be an interesting analysis to add to your manuscript.

Response:

This is a great suggestion and we are grateful to the reviewer for pointing it out. In our original submission we tested for a possible relation between test-retest reliability (ICC) and heritability at the level of individual connections. That relationship was non-significant, ruling out that the heritability estimates were related to measurement noise. However, it is still possible that the test-retest reliability increases as a function of hierarchical level (i.e. smaller differences up the hierarchy). Observing such a relationship would not only add weight to the proposed model, but also strengthen the conclusion that the overall relation between the heritability estimates and hierarchical level was

not confounded by similarly distributed noise levels (had that been the case, the test-retest reliability would have decreased with hierarchical level).

We therefore conducted the suggested analysis, which revealed that the test-retest reliability between the functional connectivity estimates of session days 1 and 2 increased up visual hierarchy ($t_{44} = 2.72$, $p = 9.2 \times 10^{-3}$), as is indeed predicted by the model. We now report this result in the context of the proposed theoretical model in the discussion of the manuscript, as well as in the new Figure 3.

In the context of this comment it is also worth noting the addition of a new analysis suggested by reviewer 2, the results of which are now presented in the same paragraph of the discussion and Figure 4. This analysis confirmed the prediction that the heritability of purely anatomical phenotypes, which we would presume are free of short-term plasticity, exhibits a monotonically decreasing relationship with hierarchical level. We believe that these additional analyses have significantly strengthened the arguments we are making in this submission.

Reviewer #2 (Remarks to the Author):

The key idea of this letter is to quantify plasticity as the Complement of the Heritability of Functional Connectivity (CHFC). This is an important and natural idea, but it raises the question -discussed by the authors- of whether functional connectivity (which relates only to correlation) is the right or the only measure that should be considered to assess plasticity in this specific dataset. As mentioned, plasticity may also be assessed using phenotypic variation with time, or longitudinal changes in brain structure. It would be interesting - and not too difficult - to evaluate if anatomical phenotypes (e.g. grey matter volume or surface area) concur with the proposed theory.

Response:

We thank the reviewer for the suggestion to evaluate if anatomical phenotypes concur with the proposed theory. This is indeed a great suggestion. Presumably, anatomical phenotypes are free of short-term plastic changes and the proposed theory predicts a monotonic decrease of heritability (increase in plasticity) when the estimates are based on anatomy alone. We have addressed this comment by computing the heritability of each visual area's grey matter volume and thickness (surface area is equal across subjects under the current atlas definition of the visual areas). As predicted, the heritability of these anatomical phenotypes exhibited a monotonic decrease (increase in plasticity) up the hierarchy. We now report the results of these new analyses in the discussion and Figure 4. Additional methodological details can be found in the new Methods section under “Two-component model of plasticity” and “Heritability of anatomical phenotypes”.

The methodological caveat that heritability, defined as the proportion of variance explained by genotypic information over the total phenotypic variance, is dependent on the measurement noise level is well accounted for. Functional connectivity measures could indeed vary with noise level and this can be measured with test re-test reliability, although it is not clear how many time point measures were used for this assessment. Other measures of the noise level per participant could be obtained, and the normalized effect sizes across the visual hierarchy estimated.

Response:

We thank the reviewer for their positive evaluation of how we accounted for the possible influence of measurement noise. We have now clarified the number of time points that were used for this assessment (2400 per session day) in the Methods section.

As to the comment regarding other measures of noise level: the crucial question is whether the heritability estimates were influenced by the noise level of the functional connectivity estimates. This is most straightforwardly tested by a test-retest measure like ICC on the connectivity estimates between regions, rather than other measures that only summarize localized noise estimates (such as tSNR).

To address the comment that the inter-individual differences in noise level could also be assessed as a function of hierarchical level, we now additionally show that the test-retest reliability in fact increases up the visual hierarchy (in the original manuscript, we only considered the possible relationship between ICC as a measure of subject-dependent variation in the phenotype and heritability at the level of individual connections regardless of hierarchical level). This new result not only underscores that differences in noise-level do not explain the observed relationship between heritability and hierarchical level (had that been the case, test-retest reliability would have decreased with hierarchical level), but also adds weight to the proposed model differentiating short- and long-term plastic effects (see our response to reviewer 1's first comment).

The discussion on how the overall plasticity could be divided into short term (ventral stream) and long term (dorsal stream) plasticity components is interesting and fits with some of the literature. It is indeed plausible that the relative amount of plasticity measured by CHFC may be governed by several factors. It also begs the question whether it would be possible that the higher level of hierarchy in the ventral stream could correspond to more stable integration neural codes and therefore may be less subject to the coding catastrophe theory.

Response:

We thank the reviewer for the suggestion that the increased stability in higher levels of ventral visual processing might correspond to more stable integration codes (e.g. to facilitate object recognition under various viewing conditions). We now discuss this possibility in the third paragraph of the discussion.

It might worth trying to -at least in part- test the theory in other sensory cortices (eg auditory)?

Response:

Though we share the reviewer's interest in testing the proposed theory in other sensory cortices, such investigations would presently come with substantial practical challenges that would need to be overcome first. For instance, the hierarchical organization of non-visual systems is much less understood and/or in e.g. the case of the auditory system the majority of processing stations for those systems are subcortical where the signal quality is rather poor. As such we think that testing this theory across other sensory cortices is not feasible at this stage.

Overall, I have found this a fascinating article and I congratulate the authors for their work: the estimation of plasticity they propose seems to be an excellent idea, and their explanation on how this informs the balance between plasticity and stability in the visual cortex is compelling.

Response:

We thank the reviewer for their encouraging words.

REVIEWERS' COMMENTS:

Reviewer #1 (Remarks to the Author):

The authors responded thoroughly to the reviewer comments and as such this manuscript will be a stronger contribution to the literature. I have no further major comments. One remaining small comment is that Figures 3 and 4 are interesting results that support your model, thus I think it would be worth including them in the results section and explaining them a bit more than with a single sentence in the discussion.

Response 2: Reviewer #2 (Remarks to the Author):

The authors' responses answer my questions fully. The new analyses are well done and their results appear to strengthen the theory considered. I have no further comments.